# Neuro-Symbolic Entity Alignment via Variational Inference

## Abstract

Entity alignment (EA) aims to merge two knowledge graphs (KGs) by identifying equivalent entity pairs. Existing methods can be categorized into symbolic and neural models. Symbolic models, while precise, struggle with substructure heterogeneity and sparsity, whereas neural models, although effective, generally lack interpretability and cannot handle uncertainty. We propose NeuSymEA, a probabilistic neuro-symbolic framework that combines the strengths of both methods. NeuSymEA models the joint probability of all possible pairs' truth scores in a Markov random field, regulated by a set of rules, and optimizes it with the variational EM algorithm. In the E-step, a neural model parameterizes the truth score distributions and infers missing alignments. In the M-step, the rule weights are updated based on the observed and inferred alignments. To facilitate interpretability, we further design a path-ranking-based explainer upon this framework that generates supporting rules for the inferred alignments. Experiments on benchmarks demonstrate that NeuSymEA not only significantly outperforms baselines in terms of effectiveness and robustness, but also provides interpretable results.

## 1 Introduction

Knowledge graphs (KGs) are crucial for organizing structured knowledge about entities and their relationships, enhancing search capabilities across various applications. They are widely used in question-answering systems (Bast & Haussmann, 2015; Dong et al., 2023), recommendation systems (Catherine & Cohen, 2016), social network analysis (Tang et al., 2008), Natural Language Processing (Weikum & Theobald, 2010), etc.. Despite their utility, real-world KGs often face issues like incompleteness, domain specificity, and language constraints, which hinder their effectiveness in cross-disciplinary or multilingual contexts. To address these issues, entity alignment (EA) aims to merge disparate KGs into a unified, comprehensive knowledge base by identifying and linking equivalent entities across different KGs. For example, aligning entities between a biomedical KG and a pharmaceutical KG allows for mining cross-discipline relationships through the aligned entities, such as identifying the same drugs and their effects on different diseases to enhance drug repurposing efforts. This alignment enables more nuanced exploration and interrogation of interconnected data, providing richer insights into how entities function across multiple domains.

Entity alignment models aim to determine the equivalence of two entities by assessing their alignment probability. Existing methods can be broadly categorized into symbolic models and neural models. Symbolic models (Suchanek et al., 2012; Jiménez-Ruiz & Cuenca Grau, 2011; Qi et al., 2021) provide interpretable and precise inference by mining ground rules, but they struggle with aligning long-tail entities, especially those without aligned neighbors. In such cases, the lack of supporting rules leads to low recall. Conversely, neural models, such as translation models (Chen et al., 2017; Sun et al., 2018) and Graph Convolutional Networks (GCNs) (Mao et al., 2020; 2021; Wang et al., 2018; Mao et al., 2022; Li et al., 2024), excel in recalling similar entities by embedding them in a continuous space, yet they often fail to distinguish entities with similar representations, causing a drop in precision as the entity pool grows. Neuro-symbolic models aim to combine the strengths of both approaches, offering the interpretability and precision of symbolic models alongside the high recall capabilities of neural models.

However, neuro-symbolic reasoning in entity alignment (EA) faces several challenges. First, combining symbolic and neural models into a unified framework is suboptimal due to the difficulty in

aligning their objectives. Current approaches either use neural models as auxiliary modules for symbolic models to measure entity similarity (Qi et al., 2021) or employ symbolic models to refine pseudo-labels (Liu et al., 2022; Chen et al., 2024b). Second, in EA task, the search space for rules is large, as the EA task requires deriving ground rules from both intra-KG and inter-KG structural patterns, leading to an exponentially large search space with increasing rule length. Finally, generating interpretations for EA remains underexplored. Effective interpretations should not only generate supporting rules but also quantify their confidence through rule weights.

To overcome these challenges, we propose NeuSymEA, a neuro-symbolic framework that combines the strengths of both symbolic and neural models. NeuSymEA models the joint probability of truth score assignment for all possible entity pairs using a Markov random field, regulated by a set of weighted rules. This joint probability is optimized via a variational EM algorithm. During the E-step, a neural model parameterizes the truth scores and infers the missing alignments. In the M-step, the rule weights are updated based on both observed and inferred alignments. To leverage long rules without suffering from the exponential search space, we employ logic deduction to decompose rules of any length into shorter, unit-length rules. This allows for efficient inference and weight updates for long rules. After training, the learned rules are adapted into an explainer, enhancing interpretability. Specifically, we reverse the logic deduction process to calculate the weights of long rules based on the learned weights of their shorter counterparts. Additionally, an explainer is introduced to efficiently generate supporting rules with quantified confidence through breadth-first search, as explanations for inferred alignments. Our contributions are summarized as below:

- **A principled neuro-symbolic reasoning framework:** NeuSymEA seamlessly integrates neural and symbolic models through a principled variational EM framework, combining their strengths for effective entity alignment.
- **Efficient optimization via logical decomposition:** We introduce a logic deduction mechanism that decomposes long rules into shorter ones, significantly reducing the complexity of rule inference and enabling efficient reasoning over large knowledge graphs.
- **Interpretable inference:** The explainer utilizes learned rules to generate support paths for interpreting both aligned and misaligned pairs. It offers two modes: **(1) Hard-anchor mode**—generates supporting paths from prealigned anchor pairs; and **(2) Soft-anchor mode**—incorporates inferred anchor pairs for more informative interpretation.
- **Empirical validation and superior results:** NeuSymEA demonstrates state-of-the-art performance on benchmark datasets, delivering both robust alignment accuracy and insightful rule-based interpretations.

## 2 Preliminaries

### 2.1 Problem statement

A knowledge graph $\mathcal{G}$ comprises a set of entities $\mathcal{E}$, a set of relations $\mathcal{R}$, and a set of relation triples $\mathcal{T}$ where each triple $(e_i, r_k, e_j) \in \mathcal{T}$ represents a directional relationship between its head entity and tail entity. Given two KGs $\mathcal{G} = \{\mathcal{E}, \mathcal{R}, \mathcal{T}\}$, $\mathcal{G}' = \{\mathcal{E}', \mathcal{R}', \mathcal{T}'\}$, and a set of observed aligned entity pairs $\mathcal{O} = \{(e_i, e_i')|e_i \in \mathcal{E}, e_i' \in \mathcal{E}'\}_{i=1}^n$, the goal of entity alignment is to infer the missing alignments by reasoning with the existing alignments. This problem can be formulated in a probabilistic way: each pair $(e, e'), e \in \mathcal{E}, e' \in \mathcal{E}'$ is associated with a binary indicator variable $\boldsymbol{v}_{(e,e')}$. $\boldsymbol{v}_{(e,e')} = 1$ means $(e, e')$ is an aligned pair, and $\boldsymbol{v}_{(e,e')} = 0$ otherwise. Given some observed alignments $\boldsymbol{v}_O = \{\boldsymbol{v}_{(e,e')} = 1\}_{(e,e') \in \mathcal{O}}$, we aim to predict the labels of the remaining hidden entity pairs $\mathcal{H} = \mathcal{E} \times \mathcal{E}' \backslash \mathcal{O}$, i.e., $\boldsymbol{v}_H = \{\boldsymbol{v}_{(e,e')}\}_{(e,e') \in \mathcal{H}}$.

### 2.2 Symbolic reasoning for entity alignment

Given an aligned pair $(e_j, e_j')$, a new aligned pair $(e_i, e_i')$ can be inferred with confidence score $w_{p,p'}$ if they are each connected to the existing pair via a relational path $p$ and $p'$ respectively, formally:

$$w_{p,p'}: \quad (e_j \equiv e_j') \wedge p(e_i, e_j) \wedge p'(e_i', e_j') \Longrightarrow (e_i \equiv e_i'), \quad (1)$$

where $p = |\mathcal{R}|^L, p' = |\mathcal{R}'|^L$ are a pair of paths each consisting of $L$ connected relations, and $w_{p,p'}$ measures the rule quality that considers the intra-KG structure and inter-KG structure, such as the

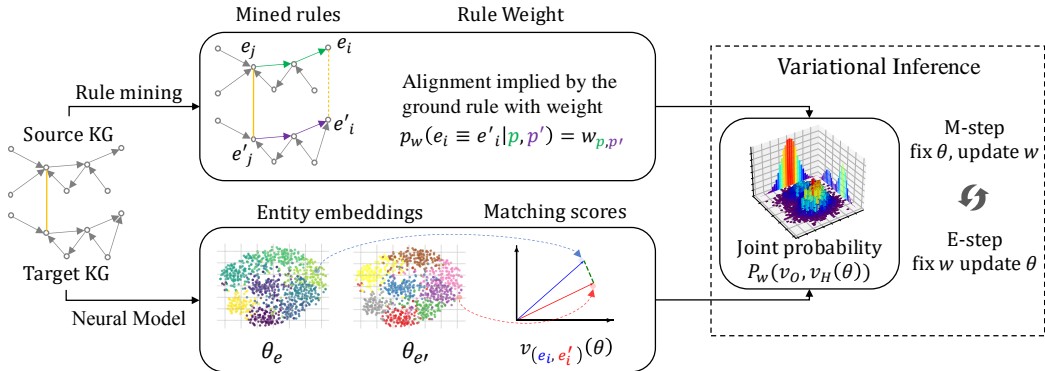

Figure 1: Framework illustration of NeuSymEA. The yellow solid line represents the alignment of an anchor pair. The symbolic model computes the matching probability of entity pairs by mining supporting rules (path pairs from anchor pairs) and evaluating their corresponding weights. The neural model learns embeddings and calculates entity-level matching scores based on embedding similarity. NeuSymEA models the agreement between the symbolic reasoning and neural representations using a joint probability distribution over observed pairs and parameterized truth scores for hidden pairs, optimized through a variational EM algorithm.

indicative of each path, and the similarity between two paths. By instantiating such *rule* with the constants (real entities and relations) in the KG pair, a symbolic model predicts the label distribution of an entity pair $(e, e')$ by:

$$p_w(\boldsymbol{v}_{(e,e')}|\mathcal{G}, \mathcal{G}'), \quad \text{for } (e, e') \in \{\mathcal{O} \cup \mathcal{H}\}. \tag{2}$$

Using logic rules to infer the alignment probability can leverage the high-order structural information for effective alignment as well as provide interpretability. However, exact inference is intractable due to the massive amount of possible instantiated rules (exponential to $L$), limiting its applicability to real-world KGs.

## 3  NEURO-SYMBOLIC REASONING FRAMEWORK FOR ENTITY ALIGNMENT

### 3.1  VARIATIONAL EM

Given a set of observed labels $\boldsymbol{v}_O$, our goal is to maximize the log-likelihood of these labels, i.e., $\log p_w(\boldsymbol{v}_O)$. Directly optimizing this objective is intractable because it requires computing an integral over all the hidden variables. Instead, we optimize the evidence lower bound (ELBO) of the log-likelihood as follows:

$$p_w(\boldsymbol{v}_O) \geq E_{q(\boldsymbol{v}_H)}[\log p_w(\boldsymbol{v}_O, \boldsymbol{v}_H) - \log q(\boldsymbol{v}_H)] = \text{ELBO}(q, \boldsymbol{v}_O; w), \tag{3}$$

here, $q(\boldsymbol{v}_H)$ is a variational distribution of the hidden variables $\boldsymbol{v}_H$. This inequality holds for all $q$ because $p_w(\boldsymbol{v}_O) = \text{ELBO}(q, \boldsymbol{v}_O; w) + D_{KL}(q(\boldsymbol{v}_H)\|p_w(\boldsymbol{v}_H \mid \boldsymbol{v}_O))$, where $D_{KL}(q(\boldsymbol{v}_H)\|p_w(\boldsymbol{v}_H \mid \boldsymbol{v}_O)) \geq 0$ is the KL-divergence between $q(\boldsymbol{v}_H)$ and $p_w(\boldsymbol{v}_H \mid \boldsymbol{v}_O)$. Under this framework, the log-likelihood $p_w(\boldsymbol{v}_O)$ can be optimized using an EM algorithm, an efficient method to find the maximum likelihood where the model depends on unobserved hidden variables: during the E-step, we fix $w$ and update the variational distribution $q$; during the M-step, we update $w$ to maximize the log-likelihood of all the entity pairs, i.e., $E_{q(\boldsymbol{v}_H)}[\log p_w(\boldsymbol{v}_O, \boldsymbol{v}_H)]$, as illustrated in Figure 1.

Explicitly representing the variational distribution $q$ is parameter intensive, which requires $\approx |\mathcal{E}||\mathcal{E}'|$ variables because the observed pairs are very sparse. To this end, we parameterize $q$ with a neural model as $q_\theta$, with $\theta$ being the parameters of the neural model.

### 3.2  E-STEP: INFERENCE

In this step, we fix $w$ and update $q_\theta$ to minimize the KL divergence $D_{KL}$. Directly minimizing the KL divergence is intractable, as it involves computing the entropy of $q_\theta$. Therefore, we follow

Qu & Tang (2019) and optimize the reverse KL divergence of $q_\theta$ and $p_w$, leading to the following objective:

$$\phi_{\boldsymbol{v}_H,\theta} = \sum_{(e,e')\in\mathcal{H}} \mathbf{E}_{p_w(\boldsymbol{v}_{(e,e')}|\boldsymbol{v}_O)} q_\theta(\boldsymbol{v}_H). \tag{4}$$

To optimize this objective, we first use the symbolic model with weighted rules to predict $p_w(\boldsymbol{v}_{(e,e')} \mid \boldsymbol{v}_O)$ for each $(e,e')\in\mathcal{H}$. If $p_w(\boldsymbol{v}_{(e,e')} \mid \boldsymbol{v}_O) > \delta$, where $\delta$ is a threshold, we treat this entity pair as a positive label; otherwise, we regard the pair as a negative pair that can be selected during negative sampling process of the neural model.

The observed labels can also be used as training data for supervised optimization. The objective is:

$$\phi_{\boldsymbol{v}_O,\theta} = \sum_{(e,e')\in\mathcal{O}} \log q_\theta(\boldsymbol{v}_{(e,e')} = 1). \tag{5}$$

The final objective for $q_\theta$ is obtained by combining these two objectives: $\phi_\theta = \phi_{\boldsymbol{v}_H,\theta} + \phi_{\boldsymbol{v}_O,\theta}$.

### 3.3 M-STEP: RULE WEIGHT UPDATE

In this step, we fix $q_\theta$ and update the rule weight $w$ to maximize $\text{ELBO}(q, \boldsymbol{v}_O; w)$. Since the right term of the ELBO in equation 3 is constant when $q_\theta$ is fixed, the objective is equivalent to maximizing the left term $E_{q_\theta(\boldsymbol{v}_H)}[\log p_w(\boldsymbol{v}_O, \boldsymbol{v}_H)]$, which is the log-likelihood function.

Specifically, we start by predicting the labels of hidden variables using the current neural model. For each $(e,e') \in \mathcal{H}$, we predict the labels $\hat{\boldsymbol{v}}_{(e,e')}(\theta)$ and obtain the prediction set $\hat{\boldsymbol{v}}_H(\theta) = \{\hat{\boldsymbol{v}}_{(e,e')}(\theta)\}_{(e,e')\in\mathcal{H}}$. In this way, maximizing the likelihood practically becomes maximizing the following objective:

$$\phi_w = \log p_w(\boldsymbol{v}_O, \hat{\boldsymbol{v}}_H(\theta)). \tag{6}$$

To obtain the pseudo-label $\hat{\boldsymbol{v}}_{(e,e')}$ using $q_\theta$, we employ the trained neural model to compute the matching score of any entity pair $(e,e') \in \mathcal{H}$. However, this strategy can easily introduce false positives into the pseudo-label set especially when the number of entities is large. To mitigate this, we consider one-to-one matching to sift only the most confident pairs. Practically, we first sort all pairs by their confidence score, then we annotate the pairs as positive following the order of the confidence. If a pair contains an entity observed in the annotated pairs, then this pair is skipped. This simple greedy strategy significantly reduces the amount of false positives.

## 4 OPTIMIZATION AND INFERENCE

### 4.1 EFFICIENT OPTIMIZATION VIA LOGICAL DEDUCTION

Inference and learning with logic rules of length $L$ can be computationally intensive, as the search space for paths grows exponentially with increasing $L$. To enhance reasoning efficiency, we decompose a rule in equation 1 using logic deduction, inspired by Cheng et al. (2023) in KG completion:

$$w_{p,p'}: \quad (e_j \equiv e'_j) \wedge \left(\bigwedge_{k=1}^{L} r_k(e_{k-1}, e_k)\right) \wedge \left(\bigwedge_{k=1}^{L} r'_k(e'_{k-1}, e'_k)\right) \implies (e_i \equiv e'_i). \tag{7}$$

Here $\bigwedge_{k=1}^{L} r_k(e_{k-1}, e_k)$ represents the path formed by $r_1, r_2, ..., r_L$ connecting $e_i$ to $e_j$ with $e_0 = e_i$ and $e_k = e_j$. This can be reorganized as *a series of single-step logic reasoning*:

$$w_{p,p'}: \quad (e_j \equiv e'_j) \wedge \left(\bigwedge_{k=1}^{L} \left[r_k(e_{k-1}, e_k) \wedge r'_k(e'_{k-1}, e'_k)\right]\right) \implies (e_i \equiv e'_i). \tag{8}$$

In this way, each logic rule of length $L$ can be viewed as *a deductive combination of $L$ short rules of length 1*. At each step, following Suchanek et al. (2012), we perform one-step inference to update $p_w(\boldsymbol{v}_{(e,e')})$ for each $(e,e') \in \mathcal{H}$ by aggregating the alignment probability from neighbors:

$$1 - \prod_{\substack{(e,r,e_t)\in\mathcal{T}, \\ (e',r',e'_t)\in\mathcal{T}'}} \left(1 - \eta(r)p_{sub}(r \subseteq r')p_w(\boldsymbol{v}_{(e_t,e'_t)})\right) \times \left(1 - \eta(r')p_{sub}(r' \subseteq r)p_w(\boldsymbol{v}_{(e_t,e'_t)})\right). \tag{9}$$

where $\eta(r)$ is a relation pattern of $r$ measuring the uniqueness of $e$ through relation $r$ given a specified tail entity $e_t$, quantified by $\eta(r) = \frac{|\{e_t|(e_h,r,e_t)\in\mathcal{T}\}|}{|\{(e_h,e_t)|(e_h,r,e_t)\in\mathcal{T}\}|}$. $p_{sub}(r \subseteq r')$ denotes the probability that relation $r$ is a subrelation of $r'$. This technique enables inference with confidence by explicitly quantifying confidence $w$ during each inference step by introducing $\eta$ and $p_{sub}(r \subseteq r')$. Moreover, in this way, the update of the weight $w$ simplifies to updating $p_{sub}(r \subseteq r')$ during the M-step (equation 6), as $\eta(r)$ for each relation $r$ is constant. In practice, the update of $p_{sub}(r \subseteq r')$ can be computed by:

$$\frac{\sum\left(1-\prod_{(e'_h,r',e'_t)\in\mathcal{T}'}\left(1-\boldsymbol{v}_{(e_h,e'_h)}\boldsymbol{v}_{(e_t,e'_t)}\right)\right)}{\sum\left(1-\prod_{e'_h,e'_t\in\mathcal{E}'}\left(1-\boldsymbol{v}_{(e_h,e'_h)}\boldsymbol{v}_{(e_t,e'_t)}\right)\right)}. \tag{10}$$

where $\boldsymbol{v}_{(e_h,e'_h)}$ and $\boldsymbol{v}_{(e_t,e'_t)}$ are labels (or pseudo-labels) from $\boldsymbol{v}_O \cup \hat{\boldsymbol{v}}_H(\theta)$.

After optimization, rule weights can be computed by taking the product of the importance scores $\eta$ of relations and the sub-relation probabilities of the corresponding relation pair:

$$w_{p,p'} := \prod_{k=1}^{L} \eta(r_k) \cdot \eta(r'_k) \cdot \frac{p_{sub}(r_k \subseteq r'_k) + p_{sub}(r'_k \subseteq r_k)}{2}. \tag{11}$$

### 4.2 Inference with interpretability

To predict new alignments, there are two approaches: using the symbolic model or the neural model. The symbolic model infers alignment probabilities with the optimized weights $w$. Due to scalability concerns, symbolic methods generally adopt a lazy inference strategy that only preserves the confident pairs implied by the neighbor structure during inference. On the other hand, the neural model computes similarity scores for all entity pairs $(e, e') \in \mathcal{H}$ using the learned parameters $\theta$, generating a ranked candidate list for each entity.

The evaluation of these models thus differs. Symbolic models are generally evaluated by precision, recall, and F1-score for their binary outputs, while neural models are assessed using hit@k and mean reciprocal ranks (MRR) for their ranked candidate lists. Following the practices in Qi et al. (2021) and Liu et al. (2022), we unify the evaluation metrics by treating the recall metric of symbolic models as equivalent to hit@1, facilitating comparison with neural models.

To enhance the interpretability of predictions, we adapt the optimized symbolic model into an explainer. For any given entity pair, the explainer generates a set of supporting rule path pairs that justify their alignment, each associated with a confidence score indicating its significance. The explainer operates in two modes: **(1) hard-anchor mode**, which generates supporting paths only from prealigned pairs, and **(2) soft-anchor mode**, which includes paths from both prealigned and inferred pairs, providing more informative interpretations.

By integrating a breadth-first search algorithm (detailed in Appendix A.3), the explainer efficiently generates high-quality interpretations. For truly aligned pairs, it typically produces high-confidence interpretations, while for non-aligned pairs, the interpretations may result in an empty set (indicating no supporting evidence) or have low confidence scores. See Figure 3 for a visualized comparison.

## 5 Experiments

### 5.1 Experimental settings

In this section, we conduct experiments to answer the following questions. **RQ1:** Can NeuSymEA outperform existing neural, symbolic, and neuro-symbolic methods in terms of alignment performance? **RQ2:** Can symbolic and embedding-based methods complement each other in our framework? **RQ3:** How does the incorporated embedding-based model affect the alignment performance? **RQ4:** How does NeuSymEA interpret the inferred entity pairs, and how is the effectiveness of interpretations with respect to the rule length?

**Datasets.** We utilize the multilingual DBP15K dataset, which consists of three cross-lingual knowledge graph (KG) pairs: ja-en, fr-en, and zh-en. Note that the original full version (Sun et al., 2017)

of this dataset contains a significant number of long-tail entities, presenting challenges for GCN-based models in terms of sparsity and large size. Therefore, many recent works (Wang et al., 2018; Mao et al., 2020; Liu et al., 2022) employ a condensed version, where long-tail entities and their connected triples are removed. For a comprehensive evaluation, we use both versions. Detailed statistics for these datasets are provided in Appendix B.1. Each dataset is divided into training, validation, and test sets following a 5-fold cross-validation scheme, with a ratio of 2:1:7.

**Baselines and metrics.** Baseline models include six neural models – GCNAlign (Wang et al., 2018), AlignE, BootEA (Sun et al., 2018), RREA (Mao et al., 2020), Dual-AMN (Mao et al., 2021), LightEA (Mao et al., 2022), one symbolic models – PARIS (Suchanek et al., 2012), and two neuro-symbolic models – PRASE (Qi et al., 2021), EMEA (Liu et al., 2022). We use Hit@1, Hit@10, and MRR as the evaluation metrics. For PARIS and PRASE that have binary outputs, we report their recall as Hit@1, following Liu et al. (2022).

**Hyperparameters.** NeuSymEA has two key hyperparameters: the number of EM iterations and the threshold $\delta$ for selecting positive pairs from the symbolic model. We tune these hyperparameters and select the best values based on the validation set. The search space for $\delta$ is $\{0.6, 0.7, 0.8, 0.9, 0.95, 0.98, 0.99\}$, while the number of iterations is searched from 1 to 9.

## 5.2 RESULTS

### 5.2.1 COMPARISON WITH BASELINES

Table 1: Entity alignment results on DBP15K dataset. The suffixes "-D" and "-L" indicate the use of Dual-AMN and LightEA as the neural models. The results of RREA and EMEA are omitted on the full dataset due to an OOM (Out of Memory) error.

| Category | Model | ja-en | | | fr-en | | | zh-en | | |
|---|---|---|---|---|---|---|---|---|---|---|
| | | Hit@1 | Hit@10 | MRR | Hit@1 | Hit@10 | MRR | Hit@1 | Hit@10 | MRR |
| **Group 1: Results on the full DBP15K dataset** | | | | | | | | | | |
| **Neural** | GCNAlign | 0.221 | 0.461 | 0.302 | 0.205 | 0.475 | 0.295 | 0.189 | 0.438 | 0.271 |
| | BootEA | 0.454 | 0.782 | 0.564 | 0.443 | 0.799 | 0.564 | 0.486 | 0.814 | 0.600 |
| | AlignE | 0.356 | 0.715 | 0.476 | 0.346 | 0.731 | 0.475 | 0.333 | 0.690 | 0.453 |
| | Dual-AMN | 0.627 | 0.883 | 0.717 | 0.652 | 0.908 | 0.744 | 0.650 | 0.884 | 0.732 |
| | LightEA | 0.736 | 0.894 | 0.793 | 0.782 | 0.919 | 0.832 | 0.725 | 0.874 | 0.779 |
| **symbolic** | PARIS | 0.589 | - | - | 0.618 | - | - | 0.603 | - | - |
| **Neuro-Symbolic** | PRASE | 0.611 | - | - | 0.647 | - | - | 0.652 | - | - |
| **Ours** | NeuSymEA-D | **0.806** | **0.942** | **0.855** | 0.827 | **0.952** | **0.871** | **0.801** | **0.925** | **0.843** |
| | NeuSymEA-L | 0.781 | 0.907 | 0.826 | **0.834** | 0.937 | **0.871** | 0.785 | 0.894 | 0.825 |
| **Group 2: Results on the condensed DBP15K dataset** | | | | | | | | | | |
| **Neural** | GCNAlign | 0.331 | 0.662 | 0.443 | 0.325 | 0.688 | 0.446 | 0.335 | 0.653 | 0.442 |
| | BootEA | 0.530 | 0.829 | 0.631 | 0.579 | 0.872 | 0.961 | 0.575 | 0.847 | 0.668 |
| | AlignE | 0.433 | 0.783 | 0.552 | 0.457 | 0.821 | 0.580 | 0.474 | 0.806 | 0.587 |
| | RREA | 0.749 | **0.935** | 0.818 | 0.797 | **0.958** | 0.859 | 0.762 | **0.938** | 0.827 |
| | Dual-AMN | 0.750 | 0.927 | 0.815 | 0.793 | 0.954 | 0.854 | 0.756 | 0.919 | 0.816 |
| | LightEA | 0.778 | 0.911 | 0.828 | 0.827 | 0.943 | 0.830 | 0.770 | 0.894 | 0.816 |
| **symbolic** | PARIS | 0.565 | - | - | 0.584 | - | - | 0.543 | - | - |
| **Neuro-Symbolic** | PRASE | 0.580 | - | - | 0.622 | - | - | 0.593 | - | - |
| | EMEA | 0.736 | - | 0.807 | 0.773 | - | 0.841 | 0.748 | - | 0.815 |
| **Ours** | NeuSymEA-D | 0.805 | 0.930 | 0.849 | 0.835 | 0.953 | 0.879 | **0.815** | 0.926 | **0.855** |
| | NeuSymEA-L | **0.811** | 0.928 | **0.854** | **0.858** | 0.954 | **0.894** | 0.804 | 0.904 | 0.840 |

To answer **RQ1**, **RQ2** and **RQ3**, we compare NeuSymEA with baseline models on two versions of the DBP15K dataset: the full version (group1) and the condensed version (group2). Results are presented in Table 1. The results for PRASE, and EMEA on the condensed DBP15K are adopted from the original EMEA paper. We employ both Dual-AMN and LightEA as the neural models in our framework, denoted as NeuSymEA-D and NeuSymEA-L, respectively. These results lead to several key observations:

**First, NeuSymEA surpasses both symbolic and KG embedding-based models.** This can be attributed to the fact that it combines the capacity of both sides: 1) the ability to precisely infer confident pairs with rules by leveraging the multi-hop relational structures; and 2) the ability to effectively model entity representations in a unified space to jointly optimize the alignment of two KGs. NeuSymEA seamlessly combines both symbolic reasoning and neural representations in a principled variational framework, leading to improved performance.

**Second, NeuSymEA outperforms both neuro-symbolic models, PRASE and EMEA.** This improvement can be largely attributed to the model objective design in our framework. While PRASE and EMEA treat the symbolic and neural models as separate components, NeuSymEA integrates them within a principled variational EM framework, unifying their objective as the log-likelihood of the observed variables. This approach allows for the joint optimization of both components, ensuring they work together to maximize the log-likelihood. By iteratively refining both components through the EM algorithm, NeuSymEA achieves a more coherent and convergent solution, leading to superior performance, as shown by the empirical results.

**Finally, NeuSymEA demonstrates superior robustness across both versions of the DBP15K dataset.** Comparisons between two groups of results offer an interesting insight: embedding-based models experience significant performance degradation when moving from the condensed version to the full version of DBP15K (e.g., MRR of Dual-AMN decreases from 0.815 to 0.717 on ja-en), while symbolic models, in contrast, show improvements. We attribute this to two key factors: (1) Embedding-based models rely on entity-level matching, which is sensitive to dataset size. As the dataset grows, the number of similar entity embeddings increases, leading to reduced accuracy. The full version of DBP15K contains significantly more entities than the condensed version, exacerbating this effect. (2) Symbolic models, on the other hand, perform path-level matching. Their effectiveness is constrained more by substructure heterogeneity and sparsity than dataset size. The full version of DBP15K includes more relational triples, which enhances the rule-mining process, ultimately making the symbolic models more effective in this scenario. **NeuSymEA, by combining symbolic reasoning with KG embeddings, mitigates the shortcomings of both approaches, making it robust to changes in dataset scale and structure. This synergy allows NeuSymEA to consistently outperform baselines in both sparse and dense settings.**

### 5.2.2 EVOLUTION OF RULES AND EMBEDDINGS

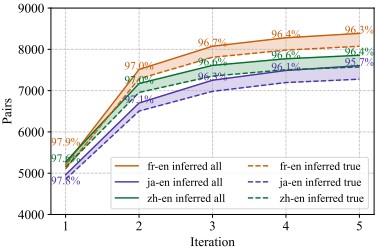 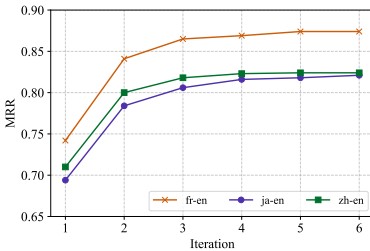

Figure 2: (Left) Evolution of rule inferred pairs, with solid lines representing total inferred pairs and dashed lines representing true inferred. The shaded areas indicate the number of false pairs. Precision values are annotated at each data point. (Right) Convergence of MRR of the neural model.

We study how rules and embeddings evolve and interact with each other during the EM steps, with results shown in Figure 2. Results in the left subplot indicate that in each EM iteration, the number of rule-inferred pairs grows consistently with high precision, implying that the embedding model continuously improves the inference performance of rules. These precise pairs, in turn, enhance the performance of the neural model. As shown in the right subfigure, the MRR of the neural model converges within a few iterations.

### 5.2.3 INTERPRETATIONS BY THE EXPLAINER

We investigate the interpretations generated by the explainer on the fr-en dataset. The left subfigure of Figure 3 shows the probability density of confidence scores for supporting rules (with a maximum rule length of 3) associated with entity pairs. Positive pairs are derived from the test set, while

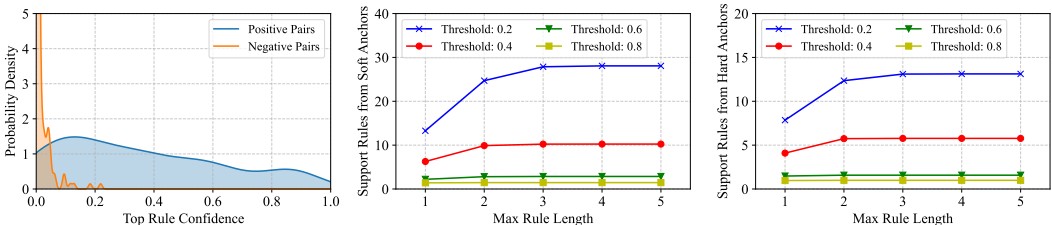

Figure 3: (Left) Probability density of the top supporting rule's confidence; (Middle) Number of supporting rules (thresholded by confidence score) relative to the maximum rule lengths under the **soft anchor mode**; (Right) Number of supporting rules relative to the maximum rule lengths under the **hard anchor mode**.

Table 2: Examples of supporting rules for query pairs in fr-en. Anchor pairs are shown in bold.

| Query Pair | Supporting Rule | Confidence |
|---|---|---|
| Maison_de_Savoie
House_of_Savoy | (Humbert_II_(roi_d'Italie), dynastie, Maison_de_Savoie), (Humbert_II_(roi_d'Italie), conjoint, **Marie-José_de_Belgique**)
(Umberto_II_of_Italy, house, House_of_Savoy), (Umberto_II_of_Italy, spouse, **Marie_José_of_Belgium**) | 0.80 |
| Légion_espagnole
Spanish_Legion | (Légion_espagnole, commandantHistorique, Francisco_Franco), (Francisco_Franco, conjoint, **Carmen_Polo**)
(Spanish_Legion, notableCommanders, Francisco_Franco), (Francisco_Franco, spouse, **Carmen_Polo,_1st_Lady_of_Meirás**) | 0.59 |
| Premier_ministre_du_Danemark
Prime_Minister_of_Denmark | (Premier_ministre_du_Danemark, titulaireActuel, **Lars_Løkke_Rasmussen**)
(Prime_Minister_of_Denmark, incumbent, **Lars_Løkke_Rasmussen**) | 0.79 |

negative pairs are created by replacing one entity in each test pair with another randomly sampled entity. The distinct confidence distributions indicate that **positive pairs generally have more evidence for alignment**, which aligns with intuition. However, the probability density distribution also reveals that some positive pairs do not have high confidence scores. Upon further examination, we found that **many test pairs are isolated, i.e., they lack directly aligned neighbors. Despite this, NeuSymEA successfully generates supporting rules for isolated pairs by exploiting multihop dependencies**. In Table 2, we provide several examples of supporting rules and their associated confidence scores for the queried entity pairs.

To examine the impact of rule length on the explainer's effectiveness, the middle and right subfigures in Figure 3 show the number of supporting rules for test positive pairs as the maximum rule length increases. Compared to hard anchor mode, the explainer in soft anchor mode generates more high-quality supporting rules by leveraging inferred pairs as complementary anchor pairs, mitigating the sparsity issue. We also observe that **increasing the maximum rule length leads to more high-quality rules; however, the number of high-confidence rules grows more slowly than lower-confidence rules.** This can be attributed to our method for calculating confidence: the logical deduction-based approach computes a rule's confidence as the product of the confidences of its decomposed length-one sub-rules (as in Equation (11)). For example, a rule with two sub-rules, each with confidence $0.8$, results in an overall confidence of $0.8 \times 0.8 = 0.64$. Considering this, the confidence score tends to decrease when the rule length increases, thus increasing the maximum length tends to discover supporting rules with lower scores.

### 5.2.4 ROBUSTNESS IN LOW RESOURCE SCENARIO

Figure 5 demonstrates the model performance under low-resource settings. As the percentage of training data decreases, all models experience noticeable drops in Hit@1 performance. Despite this, **NeuSymEA (both NeuSymEA-L and NeuSymEA-D) exhibits remarkable robustness across all datasets, consistently outperforming other models even with limited data**. Notably, with only 1% of pairs used as training data, NeuSymEA-L achieves a Hit@1 score exceeding 0.7 on fr-en, rivaling or even surpassing the performance of some state-of-the-art models trained on 20% of the data. While models such as LightEA, Dual-AMN, and BootEA show improvement with additional training data, they still fall short of NeuSymEA. In contrast, traditional models like GCNAlign and AlignE demonstrate significant struggles in low-resource settings, underscoring their heavier dependence on larger training datasets.

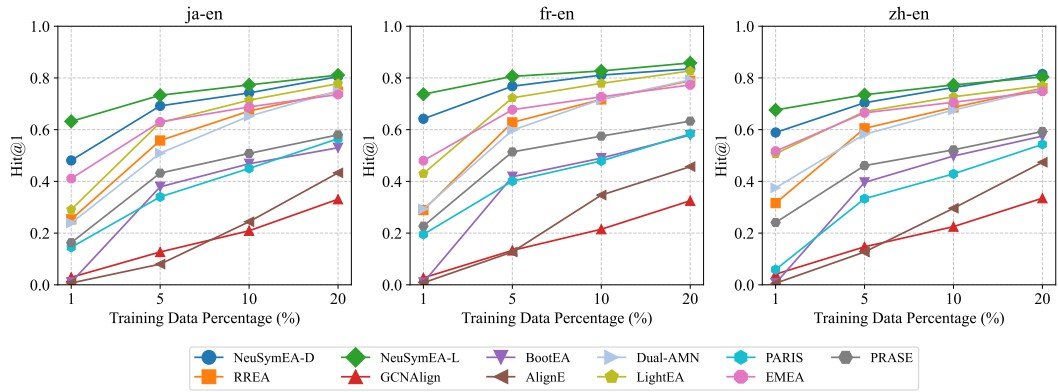

Figure 4: Alignment performance with varying percentages of pairs as training data. For brevity, we present only the Hit@1 metric, with comprehensive results available in Appendix B.2.

# 6 RELATED WORK

**Neuro-symbolic reasoning on knowledge graphs.** Neuro-symbolic methods aim to merge symbolic reasoning with neural representation learning, leveraging the precision and interpretability of symbolic approaches alongside the scalability and high recall of neural methods. In KG completion task, Guo et al. (2016) and Guo et al. (2018) employ horn rules to regularize the learning of KG embeddings; Cheng et al. (2022) and Cheng et al. (2023) model the rule-based predictions as distributions conditioned on the input relational sequences, and parameterize these distributions using a recurrent neural network; Qu et al. (2020) and Cheng et al. (2022) enhances rule grounding by augmenting the fact set using a pre-trained KG-embedding model; Qu & Tang (2019), Zhang et al. (2020) and Chen et al. (2024a) models the joint probability of neural model and the symbolic model with a Markov random field, and employ gradient descent for weight updates. Despite extensive advancements of neuro-symbolic reasoning in KG completion(DeLong et al., 2024; Cheng et al., 2024), these studies only consider single-KG structures, thus cannot be directly adopted to entity alignment which requires consideration of both intra-KG and inter-KG structures.

**Entity alignment.** Recent models have sought to combine symbolic and neural approaches for entity alignment. For instance, Qi et al. (2021) enhances probabilistic reasoning by utilizing KG embeddings, employing a KG-embedding model to measure similarities during both updating and inference processes. Liu et al. (2022) implements self-bootstrapping with pseudo-labeling in a neural framework, using rules to choose confident pseudo-labels. However, it relies solely on unit-length rules, which restricts its effectiveness for long-tail entities. Recently, Tian et al. (2024) proposes to generate interpretations to explain entity alignment, but their interpretations are subgraphs extracted by semantic matching using a pre-trained neural model.

In contrast, our work measures agreement between symbolic and neural models by modeling joint probability within a Markov random field, optimizing both components toward a unified objective within a variational EM framework. We employ logic deduction to scale reasoning with long rules of any length. These learned rules provide interpretations during inference.

# 7 CONCLUSIONS

In this work, we presented NeuSymEA, a neuro-symbolic framework for entity alignment that effectively integrates symbolic methods with neural approaches within a principled variational EM framework. By unifying these traditionally distinct methodologies, NeuSymEA addresses the challenges of substructure heterogeneity, sparsity, and uncertainty. Our empirical results demonstrate that NeuSymEA not only outperforms existing approaches but also provides interpretable alignment predictions through its path-ranking-based explainer. These findings underscore the potential of this unified neuro-symbolic framework to advance knowledge fusion by enabling effective entity alignment with uncertainty-aware interpretations.

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

# A    NOTATIONS AND ALGORITHMS

## A.1    NOTATIONS

| Notation | Description |
|----------|-------------|
| $\mathcal{G}, \mathcal{G}'$ | The source and target knowledge graphs, respectively |
| $\mathcal{E}, \mathcal{E}'$ | The sets of entities in $\mathcal{G}$ and $\mathcal{G}'$, respectively |
| $\mathcal{R}, \mathcal{R}'$ | The sets of relations in $\mathcal{G}$ and $\mathcal{G}'$, respectively |
| $\mathcal{T}, \mathcal{T}'$ | The sets of relational triplets in $\mathcal{G}$ and $\mathcal{G}'$, respectively |
| $\mathcal{O}$ | The set of observed aligned entity pairs between two knowledge graphs $\mathcal{G}$ and $\mathcal{G}'$ |
| $\mathcal{H}$ | Set of unobserved entity pairs, i.e., $\mathcal{E} \times \mathcal{E}' \backslash \mathcal{O}$ |
| $\boldsymbol{v}_{(e,e')}$ | Binary indicator variable for an entity pair $(e, e')$, where $\boldsymbol{v}_{(e,e')} = 1$ indicates alignment |
| $w_{p,p'}$ | Confidence score of a rule-inferred alignment based on paths $p$ and $p'$ |
| $p_w(\boldsymbol{v}_{(e,e')}|\mathcal{G}, \mathcal{G}')$ | Probability distribution of the alignment indicator $\boldsymbol{v}_{(e,e')}$ given knowledge graphs $\mathcal{G}$ and $\mathcal{G}'$ |
| $\theta$ | Parameters of the neural model |
| $\delta$ | Threshold to select positive pair from the symbolic model |
| $\eta(r)$ | Relation pattern measuring the uniqueness of an entity through relation $r$ |

Table 3: Notations

## A.2    COMPLEXITY ANALYSIS OF THE SYMBOLIC REASONING

In the following, we present the analysis of runtime complexity and parameter complexity one by one.

### A.2.1    RUNTIME COMPLEXITY

In variational inference, the process of learning and inferring long rules (Eq. 7) is simplified by decomposing them into unit-length rules (Eq. 8). Consequently, rule weight learning (Eq. 10) is only conducted for unit-length rules. The inference process for an $L$-length rule is then estimated by iteratively applying inference steps with unit-length rules (Eq. 9) for $L$ iterations. This strategy effectively avoids the exponential search space associated with longer rules, making the computational complexity of the inference linear with respect to the rule length $L$.

Each iteration of reasoning with unit-length rules comprises an inference step (Eq. 9) and a rule-weight learning step (Eq. 10). These steps require computing the matching probability for all possible entity pairs and relation pairs, respectively. As a result, the computational complexity of the inference step and the weight updating step are $O(|\mathcal{E}||\mathcal{E}'|)$ and $O(|\mathcal{R}||\mathcal{R}'|)$, respectively.

Thus, the total **computational complexity** for reasoning with an $L$-length rule is $O(L \cdot (|\mathcal{E}||\mathcal{E}'| + |\mathcal{R}||\mathcal{R}'|))$. Given that entity sizes are typically much larger than relation sizes, this complexity can be approximated as $O(L \cdot |\mathcal{E}||\mathcal{E}'|)$.

Notably, the computations involved in Eq. 9 and Eq. 10 can be accelerated through parallel processing, which we have implemented. This optimization reduces the **runtime complexity** to $O\left(L \cdot \frac{|\mathcal{E}||\mathcal{E}'|}{n}\right)$, where $n$ represents the number of CPU cores available for parallelization.

### A.2.2    PARAMETER COMPLEXITY

The total number of alignment probabilities for all entity pairs is $|\mathcal{E}||\mathcal{E}'|$, which is large when the entity sizes increase. We adopt a lazy inference strategy to enhance parameter efficiency. This strategy involves only saving the alignment probabilities of the most probable alignments:

$$\left\{ p_w(v_{(e_i,e_i')}), |, e_i \in \mathcal{E}, e_i' \in \mathcal{E}', p_w(v_{(e_i,e_i')}) = \max\left(\max_{e \in \mathcal{E}} p_w(v_{(e,e_i')}), \max_{e' \in \mathcal{E}'} p_w(v_{(e_i,e')})\right) \right\} \quad (12)$$

Probabilities of other entity pairs can be inferred from these saved alignment probabilities using Eq. 9. In this way, **parameter complexity** is reduced to $O(\max(|\mathcal{E}| + |\mathcal{E}'|))$.

## A.3 PSEUDO-CODE OF EXPLAINER

Below is the pseudo-code of how the explainer generates supporting rules as interpretations for the query pair. It consists of two stages: searching reachable anchor pairs, and parsing rule paths as well as calculating rule confidences.

---

**Algorithm 1** Generating Interpretations for the Queried Entity Pair with Weighted Rules

---

**Inputs:** Subrelation probabilities $p_{sub}(r \subseteq r'), p_{sub}(r' \subseteq r)$ for $r, r' \in \mathcal{R}$; Knowledge Graph pair $(\mathcal{G}, \mathcal{G}')$; Maximum rule length $\mathcal{L}$; Anchor pairs $\mathcal{A}$ with source-to-target mapping `S2T` and target-to-source mapping `T2S`; Query entity pair $(e_q, e'_q)$
**Outputs:** Ranked rules based on confidence
**1. Search Reachable Anchor Pairs within Max Depth** $\mathcal{L}$
$RN \leftarrow \text{BFS}(e_q, \mathcal{G}, \mathcal{L})$    /* Search reachable neighbors of $e_q$ using breadth-first search, max depth $\mathcal{L}$ */
$RN' \leftarrow \text{BFS}(e'_q, \mathcal{G}', \mathcal{L})$    /* Search reachable neighbors of $e'_q$ using breadth-first search, max depth $\mathcal{L}$ */
$RN_a \leftarrow RN \cup \text{T2S}(RN'; \mathcal{A})$    /* Find source nodes of reachable anchor pairs using hash mapping */
$RA \leftarrow \{(e, \text{S2T}(e; \mathcal{A})) \mid e \in RN_a\}$             /* Identify reachable anchor pairs */
**2. Parse and Rank Rules Based on Confidence**
**for** $\forall (e, e') \in RA$ **do**
    Extract paths: $p(e, e_q) = r_1 \wedge r_2 \wedge \dots, p'(e', e'_q) = r'_1 \wedge r'_2 \wedge \dots$
    **if** $|p(e, e_q)| \neq |p'(e', e'_q)|$ **then**
        $w_{p(e,e_q), p'(e',e'_q)} \leftarrow 0$              /* If path lengths don't match, rule confidence is 0 */
    **else**
        $w_{p(e,e_q), p'(e',e'_q)} \leftarrow \prod_{i=1}^{|p|} \eta(r_i) \cdot \eta(r'_i) \cdot \frac{p_{sub}(r_i \subseteq r'_i) + p_{sub}(r'_i \subseteq r_i)}{2}$    /* Compute rule confidence by products of subrelation probabilities and relation functionalities */
    **end if**
**end for**
Sort the rules $(p, p')$ by $w_{p,p'}$ in descending order
**Return** the ranked rules

---

## A.4 COMPLEXITY ANALYSIS OF THE EXPLAINER

# B EXPERIMENTAL DETAILS

## B.1 DATASET STATISTICS

The DBP15K dataset is designed for cross-lingual knowledge graph alignment and comes in two versions: the full version and the condensed version. The full version of DBP15K includes comprehensive data across three language pairs. The condensed version eliminates the less frequent entities and their connected triples. Statistics of both versions are presented below.

| Datasets | KG | Entities | Relations | Rel. Triplets | Aligned Entity Pairs |
|---|---|---|---|---|---|
| zh-en | Chinese (zh) | 66,469 | 2,830 | 153,929 | 15,000 |
|  | English (en) | 98,125 | 2,317 | 237,674 |  |
| ja-en | Japanese (ja) | 65,744 | 2,043 | 164,373 | 15,000 |
|  | English (en) | 95,680 | 2,096 | 233,319 |  |
| fr-en | French (fr) | 66,858 | 1,379 | 192,191 | 15,000 |
|  | English (en) | 105,889 | 2,209 | 278,590 |  |

Table 4: Data statistics of the full DBP15K dataset.

## B.2 COMPREHENSIVE RESULTS WITH DIFFERENT RATIO OF TRAINING DATA

Below we present the comprehensive performance evaluation with various ratios of pairs as training data. In these 3x3 subfigures, each row shows the results of the same dataset, and each column shows the results of the same metric. To summarize, we have the following observations from Figure 5:

- Neural-symbolic models, particularly our proposed NeuSymEA (NeuSymEA-L and NeuSymEA-D), consistently outperform other approaches across all datasets and metrics.

| Datasets | KG | Entities | Relations | Rel. Triplets | Aligned Entity Pairs |
|----------|-----|----------|-----------|---------------|----------------------|
| zh-en | Chinese (zh) | 19,388 | 1,701 | 70,414 | 15,000 |
|       | English (en) | 19,572 | 1,323 | 95,142 | |
| ja-en | Japanese (ja) | 19,814 | 1,299 | 77,214 | 15,000 |
|       | English (en) | 19,780 | 1,153 | 93,484 | |
| fr-en | French (fr) | 19,661 | 903 | 105,998 | 15,000 |
|       | English (en) | 19,993 | 1,208 | 115,722 | |

Table 5: Data statistics of the condensed DBP15K dataset.

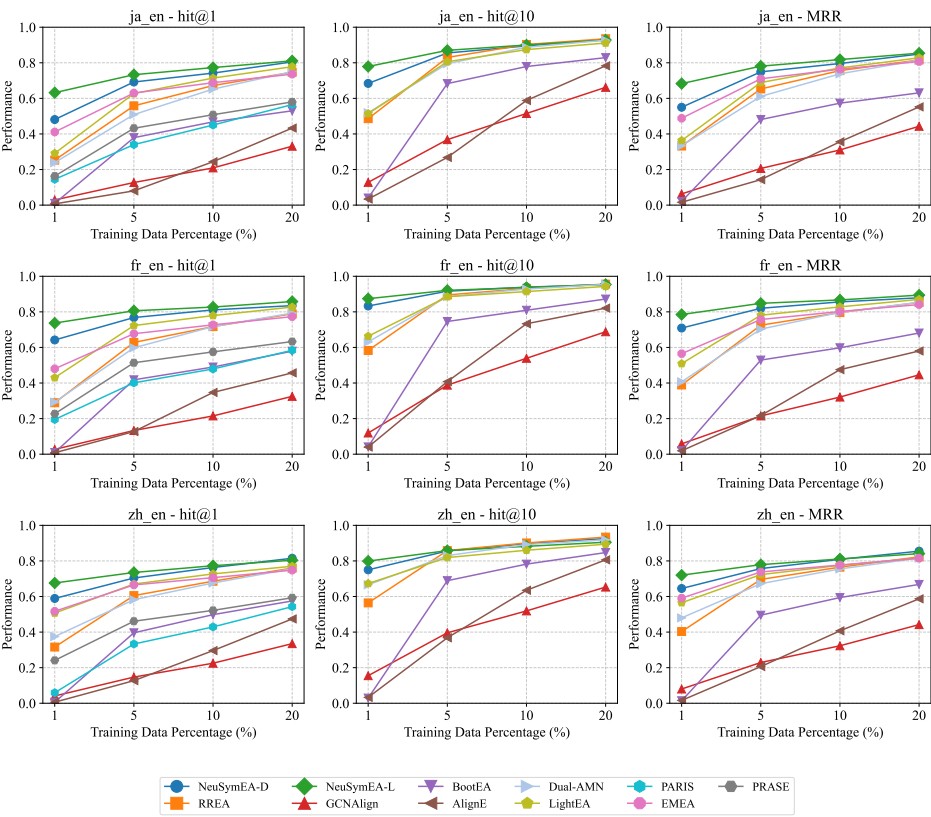

Figure 5: Alignment performance with varying percentages of pairs as training data.

These models demonstrate remarkable robustness, maintaining high performance even with limited training data (as low as 1%). Their success highlights the effectiveness of combining neural and symbolic approaches in entity alignment tasks, offering a significant advantage over traditional neural or purely symbolic methods.

- The performance of all models generally improves as the percentage of training data increases, but the rate of improvement varies significantly. Traditional neural models like GCNAlign and AlignE show the steepest improvement curves, indicating their heavy reliance on large training datasets. In contrast, neural-symbolic models, especially our proposed NeuSymEA, demonstrate high performance even with minimal training data, showcasing their efficiency and potential for low-resource scenarios.

- The experiments reveal varying levels of difficulty in entity alignment across different language pairs. Japanese-English alignment generally shows the highest performance across models, followed by French-English, while Chinese-English proves to be the most challenging. This variance underscores the importance of considering language-specific characteristics in entity alignment tasks and suggests that some language pairs may require more sophisticated approaches or additional resources for effective alignment. The ob-

served differences might be attributed to factors such as linguistic distance, writing system differences, or the availability and quality of pre-existing resources for each language pair. These findings emphasize the importance of developing flexible models that can adapt to the specific challenges posed by different language combinations.

- The exceptional performance of NeuSymEA across various scenarios – different language pairs, metrics, and data availability levels – points to their robustness and scalability. This consistency suggests that these models might be more easily adaptable to new languages or domains without requiring extensive retraining or modification. Such robustness is crucial for developing general-purpose entity alignment systems that can be deployed across diverse linguistic and domain-specific contexts. Additionally, its ability to perform well with limited data indicates potential for scalability to a wider range of languages and domains, including those with limited resources, which could significantly expand the reach and applicability of cross-lingual knowledge integration technologies.

## B.3 PARAMETER ANALYSIS

We present the hit@1 performance of NeuSymEA across three datasets, varying hyperparameters, illustrated by a three-dimensional graph. The threshold hyperparameter $\delta$ is explored within the set {0.6, 0.7, 0.8, 0.9, 0.95, 0.98, 0.99}, while the number of EM iterations ranges from 1 to 9. Performance levels are indicated using a colormap. Performance sensitivity analysis in Figure 6 reveals that for all datasets, performance generally improves as the iteration increases. On the other hand, the performance is less sensitive to the threshold $\delta$.

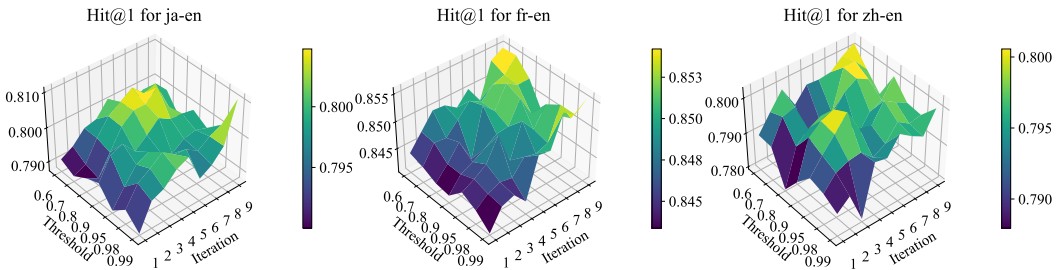

Figure 6: Performance sensitivity to hyperparameters iteration and threshold $\delta$.

