# OpenReview forum: "Neuro-symbolic Entity Alignment via Variational Inference"
_ICLR.cc/2025/Conference — Submitted to ICLR 2025_

### Official Review · Reviewer_ggkd · 2024-10-30

**Soundness:** 2
**Presentation:** 3
**Contribution:** 2
**Rating:** 6
**Confidence:** 4

**Summary:**

This paper proposes a neuro-symbolic method for entity alignment and optimizes it with the EM algorithm. The authors also design a path-ranking explainer to provide supporting rules for the predicted alignments. Experiments on the DBP15K dataset demonstrate state-of-the-art performance of the proposed method.

**Strengths:**

Leveraging the EM algorithm to alternatively learn the embedding and the path scorer is novel and interesting.

The overall writing is good.

The results of NeuSymEA compared with the baselines are promising.

**Weaknesses:**

At the beginning of the methodology section, the authors may introduce more insights into why leveraging the EM algorithm and its connection to the given problem.

The dataset is quite outdated. The authors highlight the strengths of the proposed method on long-tail entities, but DBP15K is constructed with popular entities. The authors may consider conducting experiments on OpenEA or newer datasets.

PARIS performed badly in the paper, while it has very strong performance on the OpenEA dataset. Can the authors explain this phenomenon?

Recently, there have been some methods (e.g., [1]) possessing symbolic reasoning properties and they do not require two-step optimization and the additional explainer, which seems more effective. Can the authors compare them with the proposed NeuSymEA?

[1] ASGEA: Exploiting Logic Rules from Align-Subgraphs for Entity Alignment, arXiv:2402.11000

**Questions:**

Please see Weaknesses.

---

> ### Author Response · Authors · 2024-11-21
>
> We sincerely thank the reviewer for these detailed comments that enable use to improve the quality of the work. Below we present the responses to the concerns.
>
>
>
> **[w1] Why leveraging the EM algorithm and its connection to the given problem**
>
> We aim to introduce a principled framework to combine neural and symbolic methods. At the very begining, we model the label distribution of each pair $(e, e')$ as $p_w(v_{(e, e')}|\mathcal{G}, \mathcal{G}')$. And the learning objective is to optimize maximize the log-likelihood of the observed labels: $\log p_w(v_O)$. Directly optimizing this objective is intractable because it requires computing an integral over all the hidden variables. Therefore, we resort to optimizing the evidence lower bound of $\log p_w(v_O)$:
> $$
>     p_w(v_O) \geq E_{q(v_H)} \left[ \log p_w(v_O, v_H) - \log q(v_H) \right] = \text{ELBO}(q, v_O; w),
> $$
> The insight of employing EM algorithm here is that, the ELBO contains hidden variables $v_H$. In machine learning and statistics, **EM algorithm is an efficient and scalable method to find the maximum likelihood where the model depends on unobserved hidden variables**. Thank you for this kind suggestion, we have included this insight in the lines 143-152 in the revised manuscripts, highlighted in blue.
>
>
>
> **[w2] Clarification of the dataset used**
>
> Allow us to clarify any confusion.
>
> 1. **Full version of DBP15K contains long-tail entities.** As we mentioned in the experimental setting (line269-273) and the dataset statistics (Table 4 & Table 5 at Appendix B.1), we respectifully point out that the **original full version** of DBP15K dataset does contain a significant number of **long-tail entities**, while the later **condensed version** of DBP15K only contains popular entities, which is the popularly-adopted version you mentioned. We included both versions in the main experiment (Table 1).
>
> 2. **Results on OpenEA.** OpenEA, also has two versions of dataset: the **normal version (V1)** and the **condensed version (V2)**. We include the results on both versions below. Results show that our model still outperforms the strong baselines on these datasets.
>
> - Results on OpenEA V1
>
> |   | D-W       |   | D-Y |  | EN-DE     |           | EN-FR     |           |
> | -- | - | - | - | -- | - | - | - | -- |
> |     | hit@1     | MRR       | hit@1     | MRR       | hit@1     | MRR       | hit@1     | MRR       |
> | NeuSymEA-L | **0.738** | **0.800** | **0.825** | **0.863** | **0.794** | **0.849** | **0.685** | **0.763** |
> | LightEA    | 0.727     | 0.790     | 0.817     | 0.858     | 0.780     | 0.837     | 0.672     | 0.750     |
> | Dual-AMN   | 0.715     | 0.785     | 0.817     | 0.859     | 0.786     | 0.843     | 0.649     | 0.734     |
>
> - Results on OpenEA V2
>
> |   | D-W  |   | D-Y |  | EN-DE |  | EN-FR | |
> | -| - | - | - | - | - | - | - | - |
> |  | hit@1 | MRR | hit@1 | MRR  | hit@1 | MRR  | hit@1 | MRR  |
> | NeuSymEA-L | **0.944** | **0.962** | **0.978** | **0.984** | **0.961** | **0.972** | **0.924** | **0.948** |
> | LightEA    | 0.933     | 0.954     | 0.977     | **0.984** | 0.950     | 0.963     | 0.909     | 0.939     |
> | Dual-AMN   | 0.916     | 0.945     | 0.975     | 0.981     | 0.948     | 0.964     | 0.868     | 0.914     |
>
>
>
> **[w3] Performance of PARIS**
>
> The low performance of PARIS comes from two reasons: **low recall of symbolic models** and **Graph heterougeity**
>
> - **Low recall of confidently inferred pairs.** As a symbolic model, PARIS focuses on generating confident pairs and evaluates their precision and recall, unlike neural models that assess metrics like hit@k and mean reciprocal rank. We **unify the metric and treat the recall of PARIS as hit@1, following the practice in EMEA[1]**, as stated in lines 280-281 in the manuscript. However, as a rule-based model, PARIS achieves high precision but generally suffers from low recall. For example, the precision and recall of PARIS on $DBP15K_{fr_en}$ and $OpenEA_{EN_FR}$ are:
>
> |  | Precision | Recall |
> | -- | - | -- |
> | $DBP15K_{fr\_en}$  | 0.888     | 0.584  |
> | $OpenEA_{EN\_FR} $ | 0.928     | 0.634  |
>
> As results show, the recall of PARIS on both datasets are low, although they have a high precision. Note that NeuSymEA can also generate confidence pairs with symbolic reasoning, we present the results of NeuSymEA below. Under this evaluation metric, NeuSymEA still outperforms PARIS on both datasets.
>
> |                    | Precision | Recall |
> | --| -- | - |
> | $DBP15K_{fr\_en}$  | 0.968     | 0.753  |
> | $OpenEA_{EN\_FR} $ | 0.981     | 0.803  |
>
> - **Graph Heterougeity.** OpenEA contains two mono-lingual datasets, D-W-15K and D-Y-15K. These mono-lingual datasets usually share more structural and semantic similarities compared to cross-lingual datasets in DBP15K, thus is easier to be aligned. For instance, the source and target KGs of D-Y-15K are respectively constructed from Dbpedia and YAGO, both partially use Wikipedia, leading to **overlapping structures and relations**.

---

> > ### Author Response · Authors · 2024-11-21
> > **Response the the 4rd weakness**
> >
> > **[w4] Comparison with ASGEA**
> >
> > We compare NeuSymEA with ASGEA in terms of **model design considerations** and **empirical results**.
> >
> > **1.Model design comparison**
> >
> > Both NeuSymEA and ASGEA utilize rules (relational path pairs) as interpretations. However, their model design considerations differ significantly.
> >
> > - **NeuSymEA** dynamically performs rule learning during reasoning, enabling real-time rule refinement for entity alignment. Rule quality is enhanced by the probabilistic reasoning of the symbolic model and further augmented by confidently inferred pairs from the neural model.
> > - **ASGEA**, in contrast, identifies the rule set prior to reasoning. These pre-identified rules are used to construct a subgraph for each entity, and entity alignment is then performed using a rule-informed graph neural network.
> >
> > **2.Empirical comparison**
> >
> > We compare their performance on the condensed DBP15K dataset with 30% aligned pairs used as training data.
> >
> > |            | zh-en |       | ja-en |       | fr-en |       |
> > | ---------- | ----- | ----- | ----- | ----- | ----- | ----- |
> > |            | Hit@1 | MRR   | Hit@1 | MRR   | Hit@1 | MRR   |
> > | NeuSymEA-L | 0.820 | 0.854 | 0.828 | 0.868 | 0.864 | 0.898 |
> > | ASGEA      | 0.560 | 0.660 | 0.595 | 0.690 | 0.653 | 0.745 |
> >
> > **Experimental results show that NeuSymEA outperforms the ASGEA by a large margin.**
> >
> > Note that ASGEA also has a **multimodal** setting (ASGEA-MM) where their performance is comparable to or slightly outperforms NeuSymEA.  We clarify that this setting uses textual attributes and images as **additional information**, thus direct comparing ASGEA-MM with NeuSymEA can result in **unfair comparison**.
> >
> > We appreciate your detailed and insightful comments and hope our response sufficiently addresses your concerns.
> >
> >
> > - [1] B. Liu, et al. Guiding neural entity alignment with compatibility. EMNLP 2022.

---

> ### Author Response · Authors · 2024-11-25
> **To Reviewer ggkd**
>
> Dear Reviewer,
>
> We sincerely appreciate the time and effort you have invested in assessing our work. As the deadline is approaching, we would like to kindly inquire if you have had a chance to review our rebuttal. Should there be any remaining concerns, we would be glad to provide additional clarification.
>
> We would greatly appreciate your reconsideration of our paper in light of the revisions we have made.
>
> Bests,
>
> The Authors

---

> ### Author Response · Authors · 2024-12-01
> **Summary of Responses**
>
> Dear Reviewer ggkd,
>
> We sincerely appreciate the time and effort you have invested in reviewing our paper. We understand you may have a tight schedule, so we’d like to provide a brief summary of our responses to facilitate your review:
>
> - **Insights of employing the EM Algorithm:** We clarified its role in optimizing the evidence lower bound for efficient reasoning with hidden variables. Revisions are highlighted in the manuscript (Lines 143–152).
> - **Dataset:** We clarified the DBP15K dataset statistics (full and condensed versions) and presented new results demonstrating NeuSymEA’s consistent superiority on DBP15K and newer datasets (OpenEA). Additionally, we analyzed dataset properties to contextualize the performance.
> - **PARIS Performance:** We explained its lower recall due to its symbolic nature and graph heterogeneity in DBP15K.
> - **ASGEA Comparison:** NeuSymEA is compared with ASGEA in terms of *model design considerations* and *empirical results*. NeuSymEA demonstrates stronger performance under the same setting.
>
> We hope this summary aids your review and welcome any further questions.
>
> Best regards,
> The Authors

---

> > ### Comment · Reviewer_ggkd · 2024-12-02
> >
> > Hi, thank you very much for your detailed response to my questions. I have improved my rating.

---

> > > ### Author Response · Authors · 2024-12-02
> > > **Thank you for your valuable and professional feedback**
> > >
> > > Dear Reviewer ggkd,
> > >
> > > We sincerely appreciate your thoughtful review of our paper and active engagement in discussions. Your insightful comments have been invaluable in improving our work and addressing potential misunderstandings. We are delighted that our contributions have been recognized and deeply value your professional feedback. Thank you once again for your valuable contribution.

---

### Official Review · Reviewer_jn4S · 2024-11-01

**Soundness:** 3
**Presentation:** 3
**Contribution:** 3
**Rating:** 6
**Confidence:** 4

**Summary:**

The paper presents a neuro-symbolic framework, NeuSymEA, that combines the strengths of symbolic and neural models for entity alignment in knowledge graphs. It employs a variational EM algorithm to optimize the joint probability of entity alignments, integrating embedding-based similarity and rule-based symbolic reasoning. The framework includes a path-ranking-based explainer that generates supporting rules for inferred alignments, enhancing interpretability. Experiments on DBP15K demonstrate that NeuSymEA significantly outperforms existing methods in terms of effectiveness, robustness, and interpretability.

**Strengths:**

- **Combined strengths**. The work integrates symbolic and neural models, leveraging the precision of symbolic reasoning and the high recall of neural embeddings to improve entity alignment.

- **Good performance**. The proposed framework shows superior performance and robustness across benchmark datasets, outperforming existing methods.

- **Interpretable results**. The proposed framework can provide interpretable results through a path-ranking-based explainer, enhancing the interpretability of entity alignment.

**Weaknesses:**

- **Unclear definition of rules**. According to Eq. (1), the rules used in the work are not horn rules. Instead, they are path pairs from anchor pairs. So, it would be better to provide a clear definition of the used rules.

- **Increased complexity**. The integration of symbolic and neural models with a variational EM algorithm may be complex. No discussions or experiments are provided to analyze the complexity. The framework may still face challenges with extremely large knowledge graphs due to the exponential growth of the search space for rules.

- (Minor) Missing related work: "Xiaobin Tian, Zequn Sun, Wei Hu: Generating Explanations to Understand and Repair Embedding-Based Entity Alignment. ICDE 2024: 2205-2217"

**Questions:**

-  How are the rules mined? Is it done using the algorithm described in Appendix A.2? What is the complexity of generating supporting rules?

---

> ### Author Response · Authors · 2024-11-21
> **Author Response**
>
> We sincerely appreciate your detailed and expertized comments. Below, we outline our responses to the concerns.
>
> **[w1] Unclear definition of rules**.
>
> Thank you for this detailed and constructive comment, we will provide a clear definition for the rules in the draft revision.
>
>
>
> **[w2] Complexity of EM algorithm**
>
> The EM algorithm is an iterative method for finding maximum likelihood estimates of parameters, with each iteration consisting of an E-step and an M-step. In this work, the **overall run-time complexity** of the EM algorithm is expressed as:
>
> $$n\times(C_E+C_M)$$
>
> Here, $C_E$ and $C_M$ represent the complexities of the E-step and M-step, respectively, and $n$ is the number of EM iterations. In this work, we set $n=5$ as the default, which we demonstrate to be sufficient for convergence in Figure 2 of the manuscript.
>
> For the **per-iteration complexity**:
>
> - **E-step ($C_E$)** involves parameter learning for the neural model. The complexity depends on the neural model used and can be very fast and scalable with GPU acceleration.
> - **M-step ($C_M$)** encompasses the symbolic model's reasoning process, which has been designed to be both scalable and efficient. Detailed explanations for the symbolic reasoning process can be found in Appendix A.2 of the updated manuscript. There, we outline how reasoning is accelerated using parallel computing and made parameter-efficient via lazy inference.
>
> We attach the runtime of neusymEA, and its memory consumption during symbolic reasoning below.
>
> | Run-time | Memory | GPU Memory |
> | -------- | ------ | ---------- |
> | 15min    | 868MB  | 4.33GB     |
>
>
>
> **[w2, Q1] Rule mining and complexity.**
>
> We clarify that, rule mining are performed during both **the variational EM process** and  **the interpretation generation process** but conducted differently. Below, we provide analyses for each case:
>
> **1.Rule-Based Reasoning During the Variational EM Process**
>
> In this case, long rules are not mined explicitly, we instead approximate the long rule reasoning with iterative reasoning using unit-length rules. The **parameter complexity** is $O(\max\left(|\mathcal{E}|+|\mathcal{E}'|)\right)$. The **run time complexity** is $O\left(L \cdot \frac{|\mathcal{E}||\mathcal{E}^{\prime}|}{n}\right)$, where $n$ denotes the number of CPU cores available for **parallel** computing. The detailed analysis is presented in Appendix A.2 in the updated manuscripts, highlighted in blue.
>
> **2.Rule Mining During the Interpretation Generation Process**
>
> In this case, rules are mined explictly using the algorithm described in the pseudo-code in Appendix A.3 in the updated manuscripts. The computation is dominated by the breadth-first search algoithm during searching reachable anchor pairs within max depth $L$, where the **run time complexity and parameter complexity** are both $O(d^L+d'^L)$, with $d$ and $d'$ denoting the average degree of the source and target knowledge graphs. Note that the **upper bound of the run time  and parameter complexity** is $O(|\mathcal{E}|+|\mathcal{T}|+|\mathcal{E}'|+|\mathcal{T}'|)$ and $O(|\mathcal{E}|+|\mathcal{E}'|)$ respectively, which happpen when  $L$ is large enough to potentially cover the entire graph.
>
>
>
> **[w3] related work**
>
> Thank you for suggesting the related work (ExEA), which will help enhance the comprehensiveness of our discussion. Below, we discuss the similarities and differences between NeuSymEA and ExEA:
>
> The suggested work is also designed to generate interpretations for the entity alignment results. However, there are differences between our work and theirs.
>
> - **Types of interpretations**: NeuSymEA generate a set of rules (path pairs),  while ExEA identifies a subgraph as a holistic interpretation for entity alignment results.
> - **Sources of interpretations**: NeuSymEA derives rules through reasoning results from the symbolic model, whereas ExEA extracts subgraphs through semantic matching with a pretrained neural model.
>
> Thank you for suggesting this work that boost the comprehensiveness of the discussion of related work. We have incorporated it into the revised manuscript (line469-471).
>
>
> We appreciate your insightful comments and hope our response satisfactorily addresses your concerns.

---

> > ### Comment · Reviewer_jn4S · 2024-11-26
> >
> > I appreciate the authors' response that addresses my concerns.

---

> > > ### Author Response · Authors · 2024-12-02
> > > **Appreciation for Your Valuable Feedback and Contributions**
> > >
> > > Dear Reviewer jn4S,
> > >
> > > We sincerely appreciate your professional feedback, which has helped us clarify potential confusions. We are delighted that our responses to your concerns have been acknowledged. Thank you for engaging in the review and discussion of our paper. We value your contribution and will incorporate the revisions into the final version if the paper is accepted.

---

### Official Review · Reviewer_bbmH · 2024-11-01

**Soundness:** 3
**Presentation:** 3
**Contribution:** 3
**Rating:** 6
**Confidence:** 2

**Summary:**

This paper proposes a neuro-symbolic framework, NeuSymEA, for entity alignment in knowledge graphs. NeuSymEA combines the interpretability of symbolic models with the high recall rate of neural models, optimizing entity alignment through a variational EM framework. Additionally, it includes an interpreter that generates rule-based explanation paths and confidence scores for alignment results. The main contributions are: the seamless integration of symbolic and neural models, enhancing efficiency and interpretability; an efficient logical reasoning mechanism that reduces computational complexity by decomposing long rules into shorter ones; improved interpretability through a path-ranking-based interpreter, increasing transparency and user comprehension; and finally, NeuSymEA's outstanding experimental performance, significantly surpassing baseline models on multilingual datasets and demonstrating robust performance in low-resource environments.

**Strengths:**

The NeuSymEA framework combines the interpretability of symbolic models with the high recall rate of neural models, optimizing entity alignment through a variational EM framework. This integration enables the model to effectively handle complex entity alignment tasks while balancing the strengths and weaknesses of both models within a unified framework.

**Weaknesses:**

The symbolic and neural models in the paper are integrated through the EM framework, but the trade-offs between the two in the optimization process are not explored in depth, especially in high-dimensional datasets. As a result, symbolic reasoning may still face efficiency issues, while the neural model tends to rely heavily on sparse data. Therefore, the handling strategies in these situations require further discussion.

**Questions:**

Currently, the model outperforms other methods in low-resource settings. However, will NeuSymEA's performance improve further with an increase in data volume? Is this improvement significantly different from existing baseline models?

---

> ### Author Response · Authors · 2024-11-21
> **Author Response**
>
> Thank you for your helpful feedback that helps us improve the draft. We answer concerns and address the three weaknesses in the following.
>
> **[w1] Trade-offs of the EM framework.**
>
> The variational EM framework is an efficient and principled approach for integrating two models, enhancing alignment effectiveness by leveraging their complementary strengths. **The trade-off is that this framework requires more training time.**
>
> Specifically, our framework is iteratively optimized using an EM algorithm, where each iteration consists of an E-step and an M-step. The overall run-time is $n*(T_E+T_M)$, where $T_E$ and $T_M$ denote the per-iteration run-time of the E-step and M-step, respectively, and $n$ is the number of EM iterations. In this work, the default number of iterations is $n=5$, which we demonstrate to be sufficient for convergence, as shown in Figure 2 of the manuscript.
>
>
>
> **[w2] Efficiency of symbolic reasoning**
>
> We respectfully point out that, as we have described in Page 4, section 4.1 in the manuscripts. **We have enable efficient optimization of the symbolic reasoning process**. For further details, we have included a complexity analysis of symbolic reasoning in Appendix A.2 of the updated manuscript, highlighted in blue.
>
>
>
> **[w3] Neural models’ performance on sparse data.**
>
> We respectfully point out that the statement “the neural model tends to rely heavily on sparse data” is not correct. In this work, the sparsity challenge refers to scenarios where some entities lack aligned neighbors, which typically occurs when knowledge graphs contain many low-degree entities.
>
>  **Our neuro-symbolic framework addresses this issues for the incorporated neural model** by augmenting it with precisely inferred new alignments from the symbolic model (as demonstrated in Figure 2). These inferred alignments serve as additional anchor pairs, effectively supporting low-degree entities.
>
>
>
> **[Q1] Will NeuSymEA's performance improve further with an increase in data volume? Is this improvement significantly different from existing baseline models?**
>
> The performance of all methods, including ours, is expected to improve with an increase in data volume. Below, we present the original results for the ja_en dataset (as shown in Figure 4):
>
> Hit@1 on ja_en, with x\% denoting the ratio of data used for training.
>
> |            | 1%    | 5%    | 10%   | 20%   | 30%   |
> | ---------- | ----- | ----- | ----- | ----- | ----- |
> | NeuSymEA-L | 0.632 | 0.733 | 0.773 | 0.811 | 0.828 |
> | DualAMN    | 0.239 | 0.509 | 0.652 | 0.75  | 0.801 |
> | LightEA    | 0.291 | 0.627 | 0.714 | 0.778 | 0.815 |
> | EMEA       | 0.411 | 0.63  | 0.688 | 0.736 | 0.771 |
>
> These results demonstrate that performance consistently improves as training data increases. However, as the data volume grows, the performance gap among models tends to narrow, since Hit@1 values for all models eventually approach 1.0 with sufficient training data. Yet, NeuSymEA consistently outperforms strong baselines with high training data volumn.
>
>
>
> We appreciate your comments that help us improve the quality of our work, and we hope our response can satisfactorily address your concerns.

---

> ### Comment · Reviewer_bbmH · 2024-11-27
> **The comment for author's reply**
>
> I appreciate the author's good response, as it has resolved my question，thanks.I will maintain my score.

---

> > ### Author Response · Authors · 2024-12-02
> > **Gratitude for Your Insightful Feedback and Support**
> >
> > Dear Reviewer bbmH,
> >
> > We are pleased to know that you have recognized our responses. Thank you for your involvement in the review process and the discussion period. Your insights have been instrumental in enhancing our work. We greatly value your input and will integrate the suggested enhancements into our manuscript if the paper is accepted.

---

### Official Review · Reviewer_1SAx · 2024-11-03

**Soundness:** 2
**Presentation:** 3
**Contribution:** 3
**Rating:** 5
**Confidence:** 3

**Summary:**

The paper introduces NeuSymEA, a neuro-symbolic framework for entity alignment (EA) across knowledge graphs (KGs). NeuSymEA addresses limitations in both symbolic and neural models by integrating them through a variational EM framework. This integration allows for better handling of substructure heterogeneity, sparsity, and uncertainty. The symbolic component uses Markov random fields and weighted rules for structured reasoning, while the neural component uses embedding-based models for high recall. The framework is optimized iteratively, updating rule weights and neural parameters. Furthermore, an explainer is introduced to provide interpretable justifications for entity alignments, making the model results transparent and understandable. The proposed method outperforms various baselines on benchmark datasets and demonstrates robustness, especially in low-resource scenarios.

**Strengths:**

1. The model achieves state-of-the-art results on entity alignment benchmark datasets, significantly improving alignment effectiveness. This validates the efficiency of the neuro-symbolic fusion approach and highlights its potential for broader applications in knowledge graph integration.
2. The inclusion of an explainer component provides rule-based interpretations for entity alignments, which is a major advantage for applications that require transparency and accountability, such as medical knowledge systems or legal databases.
3. The logical decomposition strategy reduces the computational complexity of rule-based inference, enabling the method to scale to larger knowledge graphs. This efficiency also facilitates handling long-tail entities effectively, making the approach practical for real-world, large-scale scenarios.

**Weaknesses:**

1. The descriptions of the variational EM algorithm and inference steps are dense and complex. For readers unfamiliar with probabilistic modeling, these sections may be difficult to understand. Simplifying the explanations or including more illustrative diagrams could enhance readability and comprehension.
2. While the paper claims efficiency improvements, it lacks a thorough complexity analysis. Specifically, the impact of rule length and dataset size on runtime and memory usage should be explicitly quantified to strengthen the argument for scalability and efficiency.
3. The paper does not provide an in-depth exploration of scenarios where NeuSymEA might underperform. For example, the framework could face challenges in extremely sparse or highly heterogeneous knowledge graphs, which warrants further discussion and empirical analysis.
4. The framework’s performance is quite sensitive to hyperparameters, such as the rule threshold (δ) and the number of EM iterations. A more comprehensive analysis of how these hyperparameters influence performance would be valuable for demonstrating the robustness of the method.
5. Although interpretability is a key feature of the framework, the utility of the rule-based explanations could be further validated. Conducting user studies or qualitative assessments would help to confirm whether the generated explanations are practically useful in real-world applications.

**Questions:**

See the Weaknesses.

---

> ### Author Response · Authors · 2024-11-21
> **Author Response**
>
> Thank you for the detailed comments, we answer the concerns below.
>
>
> **[w1] Paper presentation.**
>
> Thank you for your kind suggestions, we will include more diagrams to illustrate the framework and simplify the explanations.
>
> **[w2] Efficiency analysis**
>
> **Overall complexity**
>
> Our framwork is iteratively optimzed under a EM algorithm, each iteration consists of an E-step and an M-step. The overall complexity is thus $n*(C_E+C_M)$, where $C_E$ and $C_M$ is the complexity of the E-step and M-step, respetively, and $n$ is the number of EM iterations. In this work, the default number of EM-iterations is 5, and we show that this is enough for convergence in Figure 2 in the manuscripts.
>
> For the per-iteration complexity. $C_E$ involves the parameter learning of the neural model, which depends on the neural model used, can be very fast and scalable with GPUs. $C_M$ comes from the reasoning process of the symbolic model, which we have made scalable and efficient. The detailed analysis of the symbolic reasoning is below.
>
> **Complexity of symbolic reasoning**
>
> For the symbolic reasoning during the variational EM, we approximate the long rule reasoning with iterative reasoning using unit-length rules. This efficient approach eliminates the need to search long rules in exponential space, enabling the **parameter complexity** to $O(\max\left(|\mathcal{E}|+|\mathcal{E}'|)\right)$, which scales linearly with dataset size. And the **run time complexity** is $O\left(L \cdot \frac{|\mathcal{E}||\mathcal{E}^{\prime}|}{n}\right)$, where $n$ denotes the number of CPU cores available for parallel computing. A detailed indepth analysis is included in **Appendix A.2** in the updated manuscripts, highlighted in blue.
>
>
> **[w3] Performance of NeuSymEA in challenging scenarios.**
>
> Thank you for your detailed comments. We address the questions in two parts:
>
> 1. **NeuSymEA’s capability in handling sparse and heterogeneous KGs.**
>
> NeuSymEA demonstrates strong performance on the "full version" of DBP15K, as noted in lines 269–273. This dataset contains a substantial number of long-tail entities, introducing sparsity challenges, and its cross-lingual nature results in structural heterogeneity between the KG pairs. These characteristics highlight NeuSymEA's robustness in tackling such scenarios.
>
> 2. **Limitation of NeuSymEA.**
>
> NeuSymEA currently focuses on structural information and does not natively support multi-modal inputs (e.g., images or text). Extending it to multi-modal EA tasks would require incorporating representations of such modalities as entity features, enabling the neural model to train or compute feature-based matching scores with tailored components. We leave this for future work.
>
>
> **[w4] Sensitivity to hyperparameters.**
>
> We respectfully highlight that the performance is robust to the threshold $\delta$, as illustrated in Figure 6 in the appendix. Below are results for ja-en with varied hyperparameters. The results demonstrate that NeuSymEA is robust to $\delta$, generally improving in early iterations and converging within approximately five iterations.
>
> |                | iter=1 | iter=2 | iter=3 | iter=4    | iter=5    | iter=6    | iter=7    | iter=8 |
> | -------------- | ------ | ------ | ------ | --------- | --------- | --------- | --------- | ------ |
> | $\delta=$ 0.7  | 0.788  | 0.790  | 0.797  | 0.802     | 0.803     | **0.809** | 0.801     | 0.790  |
> | $\delta=$ 0.8  | 0.792  | 0.786  | 0.794  | 0.802     | 0.805     | 0.802     | **0.811** | 0.798  |
> | $\delta=$ 0.9  | 0.793  | 0.803  | 0.803  | 0.794     | 0.799     | **0.805** | 0.799     | 0.801  |
> | $\delta=$ 0.95 | 0.786  | 0.791  | 0.797  | **0.800** | 0.797     | 0.792     | 0.798     | 0.798  |
> | $\delta=$ 0.98 | 0.792  | 0.799  | 0.800  | **0.804** | 0.799     | 0.802     | 0.796     | 0.800  |
> | $\delta=$ 0.99 | 0.786  | 0.796  | 0.796  | 0.800     | **0.804** | 0.791     | 0.796     | 0.798  |
>
> **[w5] Practicability of rule-based explanations.**
>
> Thanks for acknowledging the significance of the interpretability. To facilatate the usability of the generated intepretations (supporting rules), we quantify their confidences based on reasoning results.  These confidences help prioritize the most important rules, ensuring more precise and actionable interpretations.
>
> Table 2 presents three examples of interpretations alongside their associated confidences.  Such rules are in the human-readable format and easy to understand. While further exploration through user studies could provide additional insights, this is beyond the scope of our current research and is left as future work.
>
> We appreciate your comments that help us improve the quality of our work, and we hope our response can satisfactorily address your concerns.

---

> ### Author Response · Authors · 2024-11-25
> **To Reviewer 1SAx**
>
> Dear Reviewer 1SAx,
>
> We deeply value the effort and time you have devoted to reviewing our submission. As the deadline nears, we wanted to follow up to check if you’ve had the opportunity to review our rebuttal. If there are any additional confusions or concerns, we would be more than happy to address them.
>
> We truly hope the improvements and clarifications we provided in our response help in reevaluating our work.
>
> Best regards,
> The Authors

---

> ### Comment · Reviewer_1SAx · 2024-11-26
>
> Thank you for your response. Your answer has partially resolved my doubts, but there are still many aspects of this article that require discussion, which you clarified will be addressed in future work. I maintain my current score.

---

> ### Author Response · Authors · 2024-11-27
> **Response to the feedback and remaining concerns**
>
> Dear reviewer 1SAx,
>
> Thank you for your thoughtful engagement with our paper and for your detailed comments. We greatly appreciate the time and effort you’ve dedicated to reviewing our rebuttal.
>
> Regarding your feedback on the aspects of our work outlined as future work, we would like to clarify any potential misunderstandings and address the remaining concerns.
>
> ### **1. On Extending NeuSymEA to Multi-Modal Inputs**
>
> We initially discussed the limitation of NeuSymEA in response to the original concern that *“The paper does not provide an in-depth exploration of scenarios where NeuSymEA might underperform”* (weakness 3). Specifically, we noted that the current model does not inherently support multi-modal information and only focuses on structural information. For completeness, we expand on the multi-modal scenario below:
>
> - **Discussion of potential multi-modal extensions**: While our current framework focuses on structural reasoning within knowledge graphs, we acknowledge the growing importance of integrating multi-modal information—such as text and images—into entity alignment tasks. Incorporating such modalities requires specialized representations and tailored components for feature-based matching. Although developing these components lies beyond the scope of our current research, we outline potential approaches to address this limitation in future work:
>   - **Multi-modal anchor alignment**: Pretrained models (e.g., BERT for text or CNNs for images) could be used to encode features from text and images. Similarities calculated from these features could identify anchor pairs, which would augment the training data and enhance the learning process.
>   - **Multi-modal fine-tuning**: Text/image representations encoded by pretrained models could serve as features for entities and be input into neural architectures like GNNs. Existing seed alignments could further refine these representations to improve performance in multi-modal contexts.
>
> ### **2. On User Studies for Rule-Based Explanations**
>
> - **Qualitative and quantitative analysis of rule-based explanations**: The reviewer suggested that user studies or qualitative assessments would provide valuable insights into the practical utility of rule-based explanations. While a full-scale user study is beyond the scope of this work, we currently quantify rule confidences to prioritize actionable interpretations. This aligns with the reviewer’s suggestion to make Neuro-Symbolic EA frameworks more practical for real-world applications, such as knowledge fusion. We provide an example below.
> - **Example application of rule-based explanations**: In cross-domain knowledge graph question answering (KG-QA), entity alignment plays a foundational role in fusing knowledge from different domain-specific KGs. This alignment is essential when integrating multi-hop relations spanning multiple graphs, as aligned entities act as crucial intermediate anchors that bridge these relations. To ensure the reliability of such alignments, our framework generates explanations in the form of path pairs, ranks them based on confidence scores, and selects the most reliable ones. Each selected explanation, represented as a pair of relational paths in their respective knowledge domains, provides a clear and interpretable description of the aligned entity. By offering transparent and trustworthy justifications for entity alignments, these explanations contribute to improving the accuracy and effectiveness of cross-domain question answering.

---

### Meta-Review · Area_Chair_z6ot · 2024-12-18

**Metareview:**

This paper proposes an EM algorithm combining symbolic logic reasoning with neural architectures. The proposed framework significantly enhances the performance of entity alignment by leveraging the combined power of symbolic rigorousness and neural capacity. The authors further propose decomposition strategies to reduce the rule length and an explainer to interpret the prediction. Comprehensive experiments are conducted on DBP15K (both full-version and condensed version) to demonstrate the effect of the proposed framework.

Strengths:
- The application of neural-symbolic framework on entity alignment task is reasonable and effective.
- The rule decomposition strategy is beneficial for reducing the complexity brought by the increased rule length. The added explainer is also useful in enhancing the model's interpretability.
- The experimental results on DBP15K demonstrate a clear performance gain compared with the baselines.

Weaknesses:
- The technical contribution is a bit limited. While the EM algorithm with neural-symbolic reasoning has already been proposed in Qu and Tang (2019), and Chen et al. (2024c), the adaptation of this framework in entity alignment is not very novel. I suggest the authors clearly discuss the contributions and compare with existing works.
- The paper claims interpretability, whereas human evaluation is not provided to strengthen this point.
- The experiments lack more datasets and recent baselines (as pointed out by reviewers and partially provided in the response).
- The complexity (including GPU runtime) should be incorporated (as pointed out by most reviewers).

**Additional Comments On Reviewer Discussion:**

- Most reviewers highlighted the complexity analysis which is important to demonstrate the strength. The authors have provided additional theoretical discussions showcasing the complexity. Further comparisons in terms of GPU runtime could be incorporated.
- Sensitivity to hyperparameters and relationship with data size were raised by reviewers and addressed by the authors.
- Human evaluation was raised by reviewer but not fully addressed.
- Addition of datasets and baselines was raised. The authors provided additional experiments to address the question.

Overall, the rebuttal has addressed some of the reviewers' questions, but not all.

---

### Decision · Program_Chairs · 2025-01-22

Reject